# Change in Collagen Fibril Diameter Distribution of Bovine Anterior Cruciate Ligament upon Injury Can Be Mimicked in a Nanostructured Scaffold

**DOI:** 10.3390/molecules26051204

**Published:** 2021-02-24

**Authors:** Zhuldyz Beisbayeva, Ainur Zhanbassynova, Gulzada Kulzhanova, Fariza Mukasheva, Cevat Erisken

**Affiliations:** 1Department of Chemical and Materials Engineering, School of Engineering & Digital Sciences, Nazarbayev University, 53 Kabanbay Batyr, Nur-Sultan 010000, Kazakhstan; Zhuldyz.Beisbayeva@nu.edu.kz (Z.B.); ainur.zhanbassynova@nu.edu.kz (A.Z.); fariza.mukasheva@nu.edu.kz (F.M.); 2Department of Biological Sciences, Nazarbayev University, 53 Kabanbay Batyr, Nur-Sultan 010000, Kazakhstan; gulzada.kulzhanova@nu.edu.kz

**Keywords:** injury, anterior cruciate ligament, bovine, collagen, fibril, diameter

## Abstract

More than 200,000 people are suffering from Anterior Cruciate Ligament (ACL) related injuries each year in the US. There is an unmet clinical demand for improving biological attachment between grafts and the host tissue in addition to providing mechanical support. For biological graft integration, it is important to provide a physiologically feasible environment for the host cells to enable them to perform their duties. However, behavior of cells during ACL healing and the mechanism of ACL healing is not fully understood partly due to the absence of appropriate environment to test cell behavior both in vitro and in vivo. This study aims at (i) investigating the change in fibril diameter of bovine ACL tissue upon injury and (ii) fabricating nanofiber-based scaffolds to represent the morphology and structure of healthy and injured ACL tissues. We hypothesized that distribution and mean diameter of ACL fibrils will be altered upon injury. Findings revealed that the collagen fibril diameter distribution of bovine ACL changed from bimodal to unimodal upon injury with subsequent decrease in mean diameter. Polycaprolactone (PCL) scaffold fiber diameter distribution exhibited similar bimodal and unimodal distribution behavior to qualitatively represent the cases of healthy and injured ACL, respectively. The native ACL tissue demonstrated comparable modulus values only with the aligned bimodal PCL scaffolds. There was significant difference between mechanical properties of aligned bimodal and unaligned unimodal PCL scaffolds. We believe that the results obtained from measurements of diameter of collagen fibrils of native bovine ACL tissue can serve as a benchmark for scaffold design.

## 1. Introduction

Anterior cruciate ligament (ACL) injuries have an estimated annual incidence of more than 200,000 in the United States alone [1]. ACL injuries cannot regenerate but heal slowly, resulting in scar tissue formation due to their poorly vascularized nature and frequency of motion. Unfortunately, currently available clinical procedures are not able to fully repair ACL injuries, mainly because of the absence of biological integration between grafts and the host tissue [2]. Reports reveal that females carry three to five times higher risk [3] and that more than one-third of the people with injured ligament tissue undergo reconstruction surgery each year [1], with an estimated annual cost of more than 4 billion dollars for nonsurgical management of ACL injuries, and about 3 billion dollars for surgical reconstruction of ACL [4]. Obviously, increased ACL loading stretches the collagen bundles/fibers, a major constituent of the tissue, until partial or complete ligament rupture occurs as a result of disruption of individual collagen fibrils [5,6,7]. A schematic of the ACL injury is given in Figure 1.

Structurally, ACL is a poorly vascularized connective tissue composed mainly of elastin, glycoproteins, glycosaminoglycans, and aligned collagen fibers (mostly type I) surrounded by fibroblast cells. It connects the femur and tibia with a basic function of stabilizing the knee joint through resisting anterior tibial translation and rotational loads. Duthon et al. reported on the anatomical structure and properties of the ACL in a review study [8]. Their report defines the cross-section of the human ACL as an irregular geometry, having an area of approximately 34 mm^2^ at the femur and 42 mm^2^ at the tibia junctions. ACL is composed of small entities called fascicles, which contain fibrils and fibroblasts. Fibrils with diameters of up to 500 nm are formed from subfibrils, and consequently form microfibrils.

Fibroblasts are ligament cells that produce components of the tissue extracellular matrix (ECM) and form about 20% of total tissue volume. The remaining proportion is the ECM, which in turn contains 60–80% water. Collagens, mainly types I and III, form 70–80% of the dry volume of the ligament [9]. A summary of ACL tissue composition is provided in Table 1.

ACL composition and structure are distorted upon injury, and its function is thus compromised as indicated by inferior mechanical properties. The arrangement of collagen fibers in ACL tissue plays a key role in load distribution since wavy parallel arrangement of fibrils provides mechanical resistance to tension force along the fiber axis. During tensile stretch, a small part of exerted load on ACL first flattens the crimp fibers, while application of additional load stretches the flattened fibers [8]. The mechanical properties of ACL are directly related to ultrastructure of collagen fibers, therefore an irregular arrangement of collagen fibers in ACL can lead to decreased mechanical resistance to the load, leading to ligament rupture. In this regard, diameter and density of collagens are critical attributes of ACL because they ultimately determine mechanical strength of the tissue [10]. Previously published studies about the dependence of mechanical properties on fibril diameter state that proper functioning of ACL requires multimodal distribution of fibril diameter [9]. In such an arrangement, collagen fibrils with smaller diameters are packed into the voids generated by packing of larger fibrils, thus increasing the packing density reflected into increased mechanical strength. For example, collagen fibril density of ACL was shown to decrease after injury because of a remarkable decrease of collagen fibers with larger mean diameter, which disrupts this hierarchical fibril organization [11].

The stress–strain behavior of ACL obtained under tensile loading shows a triphasic pattern consisting of (i) the toe region, (ii) the linear region, and (iii) the yield region [12]. The crimp pattern in the collagen fibrils straightens out at low strains, requiring relatively smaller forces, marking the toe region. Resistance force gradually increases in the linear region with elastic deformation. The start of permanent deformation is marked by the yield region. At this juncture, stress decreases due to the disruption of the collagen fibrils, eventually leading to ligament rupture. Table 2 shows the previously reported mechanical properties of ACL tissue for different species.

In the present study, bovine ACL tissue was used to determine the effect of injury on the mean and distribution of collagen fibril diameter. The choice of animal model in an experimental work is a critical component as it needs to possess similarities with human subjects. Bovine ACL is used in this study because cow knee represents a good match with the human knee in both size and proportion [21,22]. To evaluate the structural changes in ACL tissue upon injury, an injured bovine ACL model was created by applying tensile loading at 5 mm/min until rupture. Previously, the diameter of collagen fibrils was reported to exhibit structural changes upon injury, i.e., demonstrate a shift from bimodal to unimodal distribution with a reduced mean fibril diameter (Table 3). This change is the opposite of structural changes taking place during collagen fibril development, where characteristic unimodal distribution of pediatric/immature collagen fibrils gradually changes to bimodal distribution as the species matures [23,24].

Multiple biomaterials including poly(lactic-co-glycolic acid), poly(lactic acid) and polycaprolactone (PCL) have been earlier tested for their capacity to serve as scaffolding material for ACL tissue engineering [32,33]. Due to the ease of electrospinability [34], affordable cost, sufficiently long degradation time, its suitability in terms of mechanical properties [1] as well as being in the list of materials approved by the Food and Drug Administration, USA, to be used as materials of construction for a variety of biomedical devices, PCL was selected as the biomaterial for fabricating scaffolds. To the best of our knowledge based on intensive literature review, no prior study exists on the effect of bovine ACL injury on collagen fibril diameter distribution. In this work, ACL injuries were created using an ex vivo extensional deformation on bovine knee joints, and the collagen fibril diameter distribution and mean diameter were determined for both injured and healthy ACL tissues using TEM images. The PCL scaffolds were fabricated using electrospinning technique and evaluated for both diameter distribution and mean fiber diameter. Biomechanical properties of the healthy ACL tissues were characterized and compared with those of PCL scaffolds. It was hypothesized, based on prior findings for human, rabbit, and rat models (Table 3), that the mean diameter of collagen fibrils will be altered upon injury and that this structural change can be replicated in a synthetic biomaterial scaffold.

It is expected that utilization of bimodal and unimodal scaffolds fabricated in this study for in vitro and in vivo studies will enable the regenerative engineering community to better understand behavior of cells in cases of health and injury, leading to a significant contribution to the efforts of ACL repair and regeneration.

## 2. Results

### 2.1. Collagen Fibril Diameter of ACL

The tissue specimens were processed to yield representative cross-sections of healthy and injured ACL, which were then imaged using transmission electron microscopy (TEM). A histogram of fibril diameter distribution for healthy ACL tissue is given in Figure 2A for each specimen as shown in Figure 2A1–A3. Similarly, fibril diameter distribution for injured ACL tissue is given in Figure 2B for each specimen as shown in Figure 2B1–B3. As depicted in Figure 2, fibril diameter distribution for each specimen follows a similar pattern, i.e., bimodal for healthy and unimodal for injured ACL tissue. The TEM images of healthy ACL tissue (Figure 2A1–A3) clearly show that collagen fibrils are well organized and aligned longitudinally along the axis of ligament tissue, extending from femur to tibia, indicated by uniform circular cross-sections. On the contrary, the injured ACL tissue exhibits a disorganized structure where collagen fibrils are unaligned or aligned randomly along the axis of the tissue.

Combined distribution of collagen fibrils for healthy and injured ACL tissue is given in Figure 3. For a better qualitative observation of bimodal and unimodal distribution of fibrils, the mean frequency is plotted in a histogram (Figure 3A) and line graph (Figure 3B) format, with respective deviations from the mean indicated by error bars representing standard deviation (SD).

Obviously, healthy ACL tissue demonstrated a distribution with two peaks at 73.3 ± 11.5 and 213 ± 11.5 nm while the injured tissue had only one peak at 100 ± 20 nm. The range of collagen fibril diameter narrowed from 20–320 to 40–200 nm, with mean fibril diameter decreasing from 137.4 ± 15.9 to 92.7 ± 10.3 nm (*p* < 0.05, Figure 3C). The frequency-weighted fibril diameter also decreased from 147.6 to 102.5 nm following a similar trend. Overall, the collagen fibril diameter distribution of bovine ACL changed from a bimodal to unimodal upon injury with narrower range and decreased mean fibril diameter.

### 2.2. Fiber Diameter of PCL Scaffolds

The scaffolds were processed to yield representative cross-sections of aligned and unaligned PCL scaffolds, which were then imaged using scanning electron microscopy (SEM). A histogram of fiber diameter distribution for aligned scaffolds is given in Figure 4A, including each specimen as shown in Figure 4A1–A3.

Similarly, fiber diameter distribution for unaligned scaffolds is given in Figure 4B for each specimen as shown in Figure 4B1–B3. As depicted in Figure 4, fibril diameter distribution for each specimen follows a similar pattern, i.e., bimodal for aligned and unimodal for unaligned scaffolds representative of healthy and injured tissue, respectively. The SEM images of aligned PCL scaffolds visually show that PCL fibers are organized and aligned longitudinally along the axis of the scaffold. On the contrary, the unaligned PCL scaffold exhibits a disorganized structure where fibers are unaligned or aligned randomly.

Combined distribution of collagen fibrils for aligned and unaligned PCL scaffolds are given in Figure 5. For a better qualitative observation of bimodal and unimodal distribution of fibers, the mean frequency is also plotted in histogram (Figure 5A) and line graph (Figure 5B) format, with respective deviations from the mean indicated by error bars representing standard deviation (SD).

Obviously, aligned PCL scaffolds demonstrated a distribution with two peaks at 280 ± 40 and 507 ± 46 nm while the unaligned scaffolds had only one peak at 267 ± 46 nm. The range of PCL fiber diameter remained the same at 80–1000 nm (minor peaks were interruptedly observed at diameters up to 1500 nm but ignored due to their small contribution), with mean fiber diameter decreasing from 659 ± 32 to 370 ± 61 nm (*p* < 0.05, Figure 5C). The frequency-weighted fiber diameter decreased from 617 to 388 nm. Overall, the aligned and unaligned PCL scaffold fiber diameter distribution were found qualitatively similar to the healthy and injured bovine ACL tissue, respectively.

Comparison of diameters of aligned scaffolds with healthy ACL tissue and unaligned scaffolds with injured ACL tissue is given in Figure 6. This comparison reveals that although the diameter distributions are qualitatively similar (as shown in Figure 3A,B and Figure 5A,B), they are quantitatively different based on mean fiber diameters.

### 2.3. Biomechanical Properties of ACL Tissue and PCL Scaffolds

Biomechanical properties of healthy ACL tissue were evaluated by exposing the specimens to extensional deformation at a cross-head speed of 5 mm/min until failure. Stress–strain and load–elongation curves of the healthy ACL tissue are shown in Figure 7A. The tissue follows a typical triphase pattern with an initial toe region, followed by linear region, and finally the yield region. Apparently, the healthy ACL tissue had an ultimate stress and an ultimate strain of 31.9 ± 16.8 MPa and 45.8 ± 1.2%, respectively. The tissue exhibits a modulus of 0.8 ± 0.3 MPa as determined by the linear region of the stress–strain curve. Area under the curve defining the energy accumulated by straining the material, i.e., strain energy density, was calculated as 1140.1 ± 288.5 MPa. The tissue was deformed up to a load of 2158.4 ± 733 N when strained to an elongation of 21 mm (Figure 7A). Data for mechanical characterization of ACL is given in Appendix A.

Mechanical properties of PCL scaffold were also evaluated under the same conditions. The initial point in calculating the modulus was the origin of the stress-strain graph because there was no toe region observed for the scaffolds. The yielding point was defined as the offset stress of the sample, which was obtained by drawing a line parallel to the linear portion of the stress–strain curve starting from 0.2% of the strain. The yield stress was calculated as 1.3 MPa (0.82% strain) and 0.28 MPa (2.28% strain) for aligned and unaligned scaffolds, respectively. In addition, the proportionality limits of aligned and unaligned scaffolds were determined as 0.90 and 0.22 MPa, respectively. Stress-strain and load-elongation curves of the PCL scaffolds are shown in Figure 7B, and the values of parameters for PCL scaffolds are provided in Figure 8. It was determined that the aligned PCL scaffolds representing healthy ACL tissue had ultimate stress and ultimate strain of 3.98 ±1.36 MPa and 31.38 ± 2.03%, respectively. Similarly, the unaligned PCL scaffolds representing injured ACL tissue had ultimate stress and ultimate strain of 0.88 ±0.26 MPa and 17.35 ± 3.15%, respectively. The aligned and unaligned scaffolds had moduli of 1.37 ± 0.52 and 0.15 ± 0.10 MPa, respectively. The strain energy density was calculated as 123.6 ± 43.3 and 11.31 ± 3.86 MPa for aligned and unaligned scaffolds, respectively. The aligned and unaligned scaffolds were deformed up to a load of 2.93 ± 0.09 and 0.28 ± 0.11 N when strained to an elongation of 12.92 ± 0.84 and 7.42 ± 0.13 mm, respectively. Data for mechanical characterization of PCL scaffolds is given in Appendix A.

Comparison of the aligned/unaligned scaffolds and native ACL tissue in terms of mechanical properties (Figure 6 and Figure 7) revealed that aligned PCL scaffolds had modulus values similar to native healthy ACL tissue (1.37 ± 0.52 versus 0.8 ± 0.3 MPa, respectively), all other parameters having values significantly different from the healthy ACL tissue.

## 3. Discussion

This study investigated the mean and distribution of diameter of bovine ACL collagen fibrils before and after injury, and tensile properties of healthy ACL tissue. In addition, fiber diameter distribution and tensile properties of electrospun PCL nanofiber scaffolds were also evaluated to prove the concept of bimodal scaffold design for forthcoming regenerative engineering studies. Briefly, findings revealed that the collagen fibril diameter distribution of bovine ACL changed from bimodal to unimodal upon injury with subsequent decrease in mean diameter. In addition, PCL nanofiber scaffold fiber diameter distribution exhibited similar bimodal and unimodal distribution behavior to qualitatively represent the cases of healthy and injured ACL tissues. The modulus of aligned bimodal PCL nanofiber scaffolds was similar to that of the native ACL tissue.

The function of ACL tissue is directly related to biological and structural characteristics of its constituents. Collagen fibers of ACL, for instance, are regularly arranged bundles of fibrils in a parallel wave pattern along the longitudinal direction, which was clearly observed in the TEM images of injured ACL tissues. This organization enhances the mechanical properties of ACL by employing part of the exerted load to flatten wavy fibers [10]. Our findings demonstrated that a healthy bovine ACL tissue exhibits an organized/aligned structure of collagen fibrils, while this structure is diminished upon injury. A similar disordered arrangement of collagen fibrils was also reported for completely ruptured human ACLs [25] and rat Patellar Tendon [30]. Obviously, our findings on the organizational changes in the bovine ACL collagen fibrils after injury add to the existing pool of literature.

One interesting characteristic of tendon/ligament tissues is the reduction in mean collagen fibril diameter and a shift in fibril diameter distribution from bimodal to unimodal distribution. Such a structural variation was previously studied using different species for different tendon/ligament tissues, including human ACL [25], rabbit MCL [27], and mouse/rat PT [29,30]. Collagen fibril diameter distribution of rabbit MCL with two peaks at ~40 and ~190 nm changed to exhibit a single peak at ~50 nm. The range also changed from 20–270 to 40–70 nm. Similarly, collagen fibril diameter distribution of mouse PT with two peaks at ~45 and ~145 nm changed to exhibit a single peak at ~45 nm. The range also changed from 15–215 to 15–125 nm. For tendon/ligament tissues of other species, including human ACL, rabbit ACL, bovine ACL, mouse AT and mouse FT, the published studies reported only the mean fibril diameters and diameter ranges of the healthy tissues without specifically providing the distribution of fibril diameter of the tissue after injury [10,11,22,26,28,31]. Therefore, this current study also focused on the determination of collagen fibril diameter distribution of bovine ACL before and after injury. Results obtained reveal that the mean fibril diameter reduced significantly upon injury, exhibiting a shift in fibril diameter modality from bimodal to unimodal. A group led by H. Lu [28] previously reported on the mean fibril diameter and range as 124.1 ± 22.0 and 40–250 nm, respectively, for the healthy bovine ACL tissue. This is the only relevant study on bovine ACL found in literature, and our results for healthy ACL tissue are comparable with what they reported. More specifically, the mean fibril diameter and the diameter range of the healthy ACL tissue were measured as 137.4 ± 15.9 and 20–320 nm, respectively, in this study.

PCL scaffold fabricated using electrospinning technology qualitatively represent the healthy and injured ACL tissues. Fiber diameter distribution of aligned PCL scaffolds exhibited a bimodal characteristic representing healthy ACL tissue, while that of unaligned PCL scaffolds exhibited a unimodal distribution representing injured ACL tissue. Furthermore, the mean fiber diameter of unaligned PCL scaffolds was significantly smaller than that of aligned scaffolds, representing the decrease in collagen fibril diameter upon injury. Quantitatively, both aligned and unaligned PCL scaffolds possessed larger diameters as compared to collagen fibrils of healthy and injured ACL tissue. Specifically, the mean diameter of aligned PCL scaffolds was calculated as 659 ± 32 nm as compared to 137.4 ± 15.9 nm mean collagen fibril diameter of healthy ACL tissue. Similarly, the mean diameter of unaligned PCL scaffolds was calculated as 370 ± 61 nm as compared to 92.7 ± 10.3 nm mean collagen fibril diameter of injured ACL tissue. Obviously, electrospinning can be conveniently employed as a technique to fabricate fibers with diameters as small as native ACL tissue. There are published studies reporting scaffold fiber diameters as small as those of the native ACL tissue using a variety of polymeric materials [35,36,37].

Collagen fibril diameter and distribution are known as the determinants of tissue mechanical properties, and changes in fibril diameter and distribution were found to have a direct effect on the mechanical properties [9,10,38]. A bimodal distribution as seen in the healthy ACL tissue leads to stronger mechanical properties because the interfibril spaces between larger fibrils are filled with smaller fibrils to form a highly packed ECM structure. Upon disruption of this hierarchical structure, the capacity of ligament tissue to resist physiologic loads diminishes, leaving the tissue mechanically weaker. In this study, only the mechanical properties of healthy ACL tissue were tested. Results of tensile tests revealed that the healthy ACL tissue resisted up to a load of 2158.4 ± 733 N when strained by 45.8 ± 1.2%. Literature is scarce in terms of tensile properties of bovine ACL and the available studies report data collected at different deformation rates. Unfortunately, to the best of our knowledge, there is no report for the physiologic level deformation rate for ACL tissue. Based on our previous experience [1] and the available literature data, we preferred a cross-head speed of 5 mm/min. The most relevant study performed with bovine ACL tissue reports a stiffness of 577.3 ± 483.1 N/mm and a maximum load of 4372 ± 1485 N at 19.3 ± 18% elongation determined at 60 mm/min speed [19].

The aligned PCL scaffolds fabricated to represent healthy ACL tissue exhibited similar modulus of elasticity with native ACL tissue (1.37 ± 0.52 versus 0.8 ± 0.3 MPa, respectively, *p* > 0.05). In terms of other parameters tested in this study, such as ultimate stress, ultimate strain and energy density, the native ACL tissue was superior (*p* < 0.05) to both aligned and unaligned PCL scaffolds. A comparison within the two scaffold groups demonstrated that aligned PCL scaffolds were superior (*p* < 0.05) to their unaligned counterparts in terms of all tensile test parameters, which is also in agreement with previously published results of tensile properties of aligned and unaligned PCL scaffolds [39,40]. For a meaningful comparison of our findings with those of the others requires that the scaffolds are made of polymers of the same properties (such as molecular weight) and mechanically tested under identical conditions. Unfortunately, investigators in the field of scaffold technology use a variety of PCL from different sources and with different molecular weights. Similarly, the mechanical properties are tested under conditions of the researcher’s interest. Therefore, a meaningful comparison has not been possible. However, we earlier used PCL obtained from the same manufacturer and with the same properties to make unaligned scaffolds and tested them under identical conditions. Accordingly, we found an ultimate stress and strain values of 0.47 MPa and 350% [34], 0.81 MPa and 259% [41], 1.43 MPa and 259% [1] as compared to 0.88 ± 0.26 MPa and 17.35 ± 3.15% of this study, respectively. Apparently, the same PCL yielded similar stress and strain values except the 17.35% strain obtained in this study.

This study certainly has some methodological limitations that influenced the interpretation of the findings from this research. Firstly, the mechanical properties of injured ACL tissue could not be evaluated due to unavailability of necessary instrumentation. Such an evaluation would allow for a comparison between unaligned PCL scaffolds and injured PCL tissue in terms of mechanical properties. In addition, fiber diameter distribution of the PCL scaffolds was qualitatively similar to collagen fibril diameter distribution of the native ACL tissue. Quantitatively, both mean fiber diameter and diameter distribution of PCL scaffolds were different from ACL tissue due to some obvious factors discussed in the previous paragraphs. Therefore, in this regard, this study can be considered as a very preliminary proof of concept work and further investigation is needed to fabricate scaffolds with comparable mean diameter, diameter distribution, and mechanical properties. Furthermore, in the present study, the mechanical property of ACL tissue was tested by applying uniaxial tensile load to the femur-ACL-tibia complex, while sagittal plane (anatomical orientation) biomechanics are the major mechanism of ACL loading [5]. This point should be taken into consideration when designing experiments of similar work in the future.

## 4. Materials and Methods

This research study involves two major experimental works: (i) harvesting ACL tissues from bovine and fabricating electrospun nanofiber scaffolds, and (ii) their characterization in terms of diameter of structural components (collagen and PCL for ACL and PCL scaffold, respectively) using TEM/SEM and tensile biomechanical properties. Experimental design and procedures followed in the harvesting and characterization of ACL tissue are given in Figure 9.

### 4.1. Materials

All the chemicals were procured from Sigma Aldrich with product numbers provided below. Gluteraldehyde (2.5%, Sigma Aldrich, St. Louis, MO, USA, #G5882), osmium tetroxide (1%, Sigma Aldrich, #75633), 0.1 M phosphate buffer solution (Sigma Aldrich, #P5244), ethanol (Sigma Aldrich, #E7023), propylene oxide (Sigma Aldrich, #82320), epoxy embedding medium 812 substitute (Sigma Aldrich, #45345), epoxy embedding medium hardener DDSA (Sigma Aldrich, #45346), epoxy embedding medium hardener MNA (Sigma Aldrich, #45347), epoxy embedding medium accelerator DMP 30 (Sigma Aldrich, #45348), oil, polycaprolactone (Sigma-Aldrich, #440744), dichloromethane (Sigma Aldrich, #270997), *N*-N-Dimethylformamide (Sigma Aldrich #319937).

### 4.2. Methods

#### 4.2.1. Tissue Harvesting

The bovine knee joints were obtained from a local abattoir soon after the animal was sacrificed (*n* = 9) and tested for biomechanical properties within 3 h (*n* = 4) or after storing at −20 °C (*n* = 1). The frozen tissue was thawed at room temperature before the experiment was initiated. All tendons and ligaments of the joint other than the ACL were cleared off (Figure 9A) to prepare the ACL tissue for biomechanical characterization. More information on the tissue specimens is given in Figure 9B.

#### 4.2.2. Transmission Electron Microscopy Characterization

Collagen fibril diameter and diameter distribution were obtained from transmission electron microscopy (TEM) images of sectioned tissues. For TEM characterization of healthy ACL tissues (*n* = 3/group), location of ACL was identified, extended for better vision, and it was dissected from the bones (Figure 9C) to finally obtain a specimen with dimensions of 1 mm × 1 mm from the midsection of the tissue. Injured ACL tissues were obtained after mechanical rupture of the intact ACL tissues and a specimen with dimensions of 1 mm × 1 mm from the midsection of the tissue was used for TEM characterization.

The specimens were fixed with 2.5% solution of glutaraldehyde (Figure 9D) to prevent any possible alteration of cell structure during processing, including cell morphology, volume, and spatial configuration. During fixation, the specimens were initially kept at room temperature and then they were gradually cooled to 4 °C. This type of cooling is necessary to slow autolytic processes and reduce tissue shrinkage. The specimens were washed with phosphate buffer solution (PBS) three times, each for 10 min. To give more stability to the specimen, a secondary fixation was carried out using 1% osmium tetroxide for 2 h. Osmium tetroxide was used both as a fixative and contrasting material. After fixation, specimens were washed with PBS two times, each for 10 min. The specimens were then dehydrated through a graded series of 50% ethanol for 40 min, 70% ethanol for 12 h, 96% ethanol for 20 min two times, 100% ethanol for 15 min two times, a mixture of 100% ethanol and propylene oxide for 10 min, and propylene oxide for 15 min two times, which was used as a transitional solvent. The graded series of ethanol was used to provide smooth transitioning to avoid any alterations in tissue structure. Then, the dehydrated specimens were infiltrated using different mixtures of resin and propylene oxide. Resin was prepared from epoxy mix medium components, 812, DDSA (dodecenylsuccinic anhydride), MNA (methyl nadic anhydride), in different proportions. Infiltration was conducted to fill blocks of samples with resin in order to make samples hard enough to resist the pressure during sectioning and cutting. Samples were put into a 1:1 mixture of resin and propylene oxide for 2 h at 37 °C. Then, the mixture was changed to 3:1 proportion for 2 h at 37 °C, followed by pure resin for 12 h. The next step was embedding with different mixtures of resin and propylene oxide in molds for 24 h. Polymerization of embedded samples occurred during the next two days at 60 °C (Figure 9E).

Finally, thin sections were cut perpendicular to the ligament’s longitudinal axis using an ultramicrotome (Boeckeler PT-PC PowerTome Ultramicrotome, Boeckeler, Tucson, AZ, USA). This special tool is used for cutting sections of the specimens by movement of the block to the sides by controlled increments over a diamond knife. This device has a trough that is filled with distilled water, where the sections cut are collected. The size of each section was selected to be approximately 60 nm to get images with the best resolution.

In order to take images of ACL sections with high magnification, a transmission electron microscope (JEOL JEM-1400 Plus 120 kV TEM, JEOL, Peabody, MA, USA) was used (Figure 9F). The blocks were adjusted to optimize perpendicular cut of fibrils to obtain cross-sectional views (Figure 9G), representing specimens from different locations of the harvested tissue. Approximately 10 sections were imaged for each specimen. In this way, approximately 30 sections were obtained for each group of healthy and injured ACL tissues.

#### 4.2.3. ACL Fibril Diameter and PCL Fiber Diameter/Alignment Measurements

A total of ten parallel lines with equal distances were drawn on the TEM images and the diameter of fibrils intersecting these lines were measured. This technique was previously used to measure the fiber diameter of scaffolds imaged using scanning electron microscopy [42]. Image processing software, Image J (National Institutes of Health, Bethesda, MD, USA), was used to measure the diameter of each fibril. A minimum of 100 fibrils was measured for each section and three sections were used for each joint. More than 300 representative readings were obtained for each group of healthy and injured ACL tissues. Finally, diameter distribution, mean diameter, and range were obtained for each group (healthy and injured ACL). A similar approach was employed to measure the fiber diameter and alignment of PCL scaffolds using SEM images. Briefly, the lines were drawn on the SEM images and the fibers intersecting these lines were evaluated using ImageJ. The diameter of PCL fibers was measured as described above and the alignment (*n* = 5) was obtained by measuring the angle of the fiber with reference to the horizontal plane.

#### 4.2.4. Nanofiber Scaffold Fabrication with Electrospinning

The scaffolds were fabricated from PCL using an electrospinning process. The PCL (#440744) was obtained from Sigma Aldrich and had an average molecular number of 80,000, as provided by the manufacturer. The electrospinning instrument was procured from Inovenso (NE300 Laboratory Scale, Istanbul, Turkey).

PCL solutions with two different concentrations (20% *w*/*v* and 40% *w*/*v*) were prepared for electrospinning. Briefly, 2 g of PCL was dissolved in a 10 mL mixture of DCM and DMF (4 mL DCM and 6 mL DMF) to prepare a solution of 20% PCL. PCL solution with the concentration of 40% *w*/*v* was prepared by dissolving 4 g of PCL in a 10 mL mixture of DCM and DMF at 1:1 ratio. The mixtures prepared were stirred gently for 12–16 h using a magnetic stirrer to ensure homogeneity.

Aligned scaffolds with bimodal fiber distributions were fabricated by co-electrospinning of PCL with 20% and 40% concentrations on a rotating drum (diameter = 10 cm). The solutions were transferred to separate syringes facing the drum collector. Solution with 20% PCL was fed at a rate of 0.1 mL/h, while solution with 40% PCL was pumped at 0.25 mL/h rate. Electrospinning was performed in a room where temperature and humidity were controlled by an air conditioning system available in the room. Temperature and humidity in the room were kept around 23 °C and 50%, respectively. The fibers were collected at 10 kV on the drum rotating at 2000 rpm (10.47 m/s) located at a 12 cm distance. The unaligned fibers were generated by electrospinning the 20% PCL solution at a flow rate of 0.25 mL/h at 10 kV and a distance of 12 cm onto a stationary collector.

#### 4.2.5. Mechanical Characterization under Tension and Creation of Injured ACL Tissues

Biomechanical properties of ACL tissues were measured using a uniaxial material testing machine (Tinius Olsen H25 ST, Horsham, PA, USA) equipped with a 5 kN load cell. This system allows for achieving an accuracy of better than 0.2% of the reading from 0.2–100% of the load cell capacity as reported by the manufacturer. The tibia–ACL–femur joints were first analyzed for their dimensions, including length, thickness, and width, which were used for evaluating biomechanical properties. The measurements were made using a standard digital caliper. The ACL was gently squeezed between the jaws of the caliper along its width and its thickness recorded. At this thickness, when it is in the squeezed position, the width was measured using another caliper. At least three measurements were made for each dimension. The joint was then attached to the testing unit using a custom-made jaw assembly (Figure 9H) and positioned such that axis of the ACL was colinear with the load axis of the testing device. The specimens were stretched at a constant cross-head speed of 5 mm/min until failure (Figure 9I–L). A total of five joints (*n* = 5) were used to test biomechanical properties. The tensile tests enabled us to create injured ACL tissues and three out of five specimens were employed in fibril diameter measurements representing injured ACL tissue. Data were recorded as load versus displacement and then converted to stress versus strain and presented accordingly.

Mechanical properties of PCL scaffolds were measured using a uniaxial material testing machine (MTS Criterion Model 43, MTS Systems Co., Eden Prairie, MN, USA) equipped with a 1 kN load cell. The device works with an accuracy of ±0.5% of the force applied and ±1% of the set speed as reported by the manufacturer. In addition, the load cell (MTS 661.18 Force Transducers) operates with a repeatability of 0.03% of the full scale. The scaffold specimens were cut in rectangular shapes with a length of 5 cm and width of 1 cm. Their thicknesses were then measured using a standard digital caliper. At least three specimens were used for dimensional analysis. Briefly, first the caliper was calibrated based on the thickness of two microscope slides. The mat was inserted between two microscope slides glass and its thickness measured. Use of microscope slides was needed to avoid excessive pressure on the electrospun mats. For mechanical tests, the scaffolds were fixed using custom-made jaws and positioned such that the gauge length was 3 cm and that the scaffolds were strained in the direction of fiber alignment. In the experiment, the samples were first stretched to an initial load of 0.03 N to straighten the mats, and the experiment was then initiated. The specimens were stretched at a constant cross-head speed of 5 mm/min until failure. At least three specimens were used to test mechanical properties of each scaffold group. For statistical purposes, employment of three scaffolds per group was previously tested and validated to be reliable to draw conclusions [1,41]. Data were recorded as load versus displacement and then converted to stress versus strain and presented accordingly. Briefly, the cross-sectional areas of the scaffolds were calculated and the load was normalized by this area to obtain corresponding stress values. The strain was calculated by normalizing the change in length by the gauge length and converted to percent wherever applicable.

#### 4.2.6. Scanning Electron Microscopy Characterization

The scaffolds were first coated with gold with a thickness of 5 nm at a current of 20 mA using a turbo-pumped sputter coater (Quotrum Q150T ES, UK). They were then imaged using SEM (Crossbeam 540, Zeiss Gemini 2, ZEISS, Oberkochen, Germany) with magnifications from 3.5 to 9 K.

#### 4.2.7. Statistical Analysis

Comparison of fibril diameter of healthy and injured ACL tissues, diameter of fibers of PCL scaffolds representative of healthy and injured ACL tissues, and comparison between tensile properties of aligned and unaligned scaffolds were all analyzed using an unpaired Student’s *t*-test. Mechanical properties of scaffolds and native ACL tissue were compared using one-way analysis of variance (ANOVA). The difference was considered significant for *p* < 0.05.

## 5. Conclusions

In the current research, we reported the results of a preliminary study on the morphological and structural characteristics of native bovine ACL tissue including collagen fibril organization, mean diameter, and diameter distribution. Nanofiber scaffolds representing structural properties of both healthy and injured ACL tissues were formed using electrospinning. Findings revealed that the collagen fibril diameter distribution of bovine ACL changed from bimodal to unimodal upon injury with subsequent decrease in mean diameter. PCL scaffold fiber diameter distribution exhibited similar bimodal and unimodal distributions to qualitatively represent the cases of healthy and injured ACL tissues. The native ACL tissue demonstrated comparable modulus values only with the aligned bimodal PCL scaffolds. A comparison of mechanical properties of aligned bimodal and unaligned unimodal PCL scaffolds yielded significant difference between the two groups. Currently, no data are available for the fabrication and application of nanofibrous scaffolds possessing bimodal distribution for ACL regeneration. Such a scaffold design for ACL repair and regeneration, proposed here for the first time, is a deviation from the conventional unimodal approach. Findings of this study will be used as an input for the upcoming in vitro and in vivo studies and are, therefore, expected to have significant impact on the efforts of orthopedic research community to solve an important societal and economic healthcare problem.

## Figures and Tables

**Figure 1 molecules-26-01204-f001:**
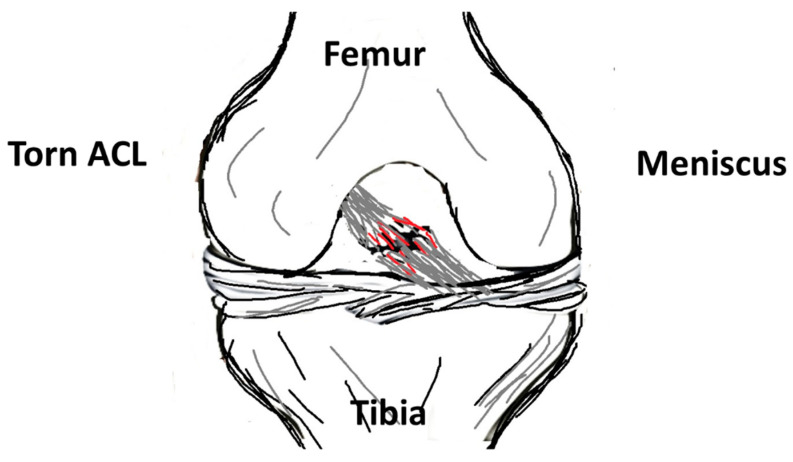
Schematic of a torn/ruptured anterior cruciate ligament (ACL) tissue.

**Figure 2 molecules-26-01204-f002:**
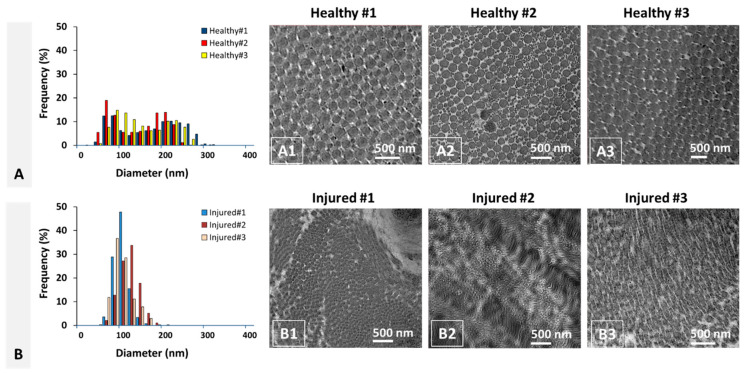
Diameter distribution of collagen fibrils from (**A**) healthy and (**B**) injured ACL tissue and respective (**A1**–**A3**,**B1**–**B3**) transmission electron microscopy (TEM) images. (**A1**–**A3**) correspond to specimens obtained from ACL tissue of each healthy subject (*n* = 3). (**B1**–**B3**) correspond to specimens obtained from ACL tissue of each injured subject (*n* = 3).

**Figure 3 molecules-26-01204-f003:**
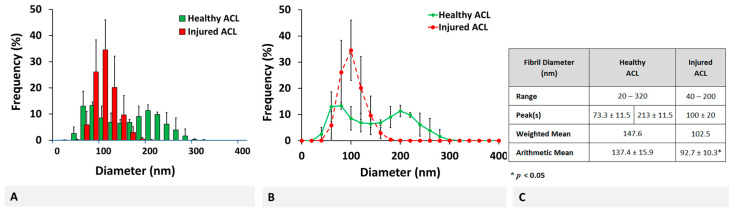
Combined healthy and injured collagen fibril diameter distributions in a (**A**) histogram and (**B**) line graph format, together with (**C**) descriptive statistics. * Indicates statistically significant difference at *p* < 0.05 (*n* = 3) and error bars represent SD.

**Figure 4 molecules-26-01204-f004:**
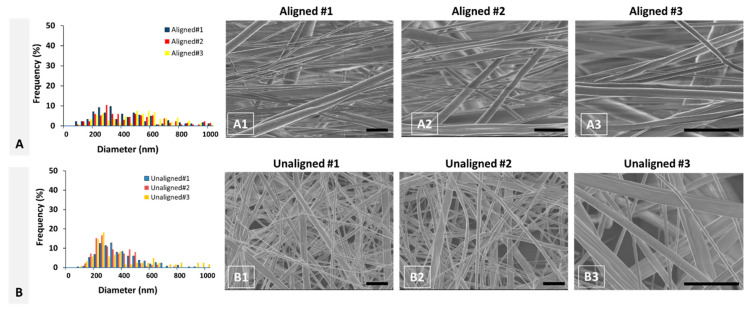
Diameter distribution of (**A**) aligned and (**B**) unaligned PCL fibers and respective (**A1**–**A3**,**B1**–**B3**) scanning electron microscopy (SEM) images. (**A1**–**A3**) correspond to individual specimens obtained from aligned PCL scaffolds (*n* = 3). (**B1**–**B3**) correspond to individual specimens obtained from unaligned PCL scaffolds (*n* = 3). Scale bar = 4 μm.

**Figure 5 molecules-26-01204-f005:**
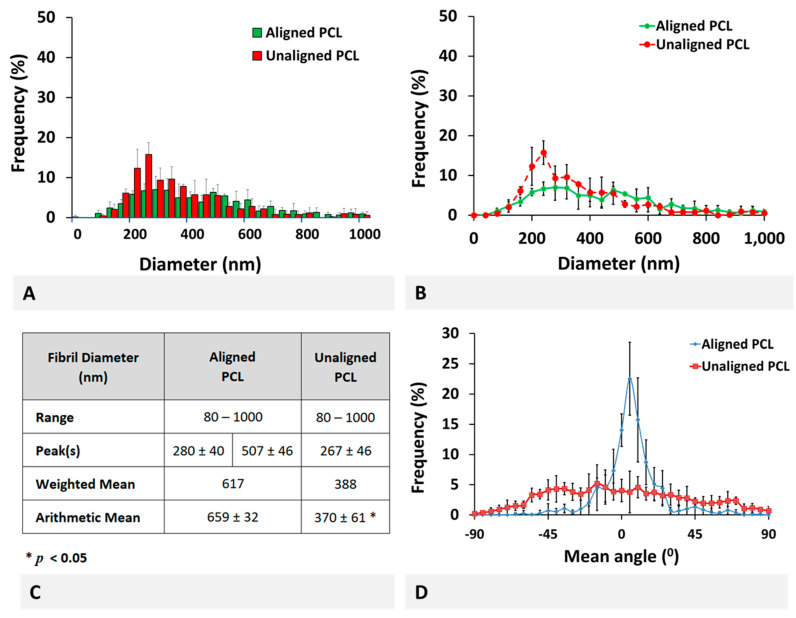
Combined aligned and unaligned polycaprolactone (PCL) fiber diameter distributions in (**A**) histogram, (**B**) line graph, and (**C**) descriptive statistics format, together with alignment measurements (**D**). * Indicates statistically significant difference at *p* < 0.05 (*n* = 3 for diameter measurements and *n* = 5 for alignment measurements) and error bars represent SD.

**Figure 6 molecules-26-01204-f006:**
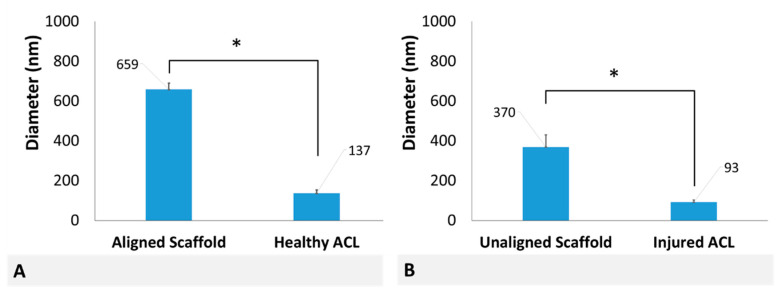
Comparison of mean diameters of (**A**) aligned scaffolds with healthy ACL and (**B**) unaligned scaffolds and injured ACL. * Indicates statistically significant difference at *p* < 0.05 (*n* = 3) and error bars represent SD.

**Figure 7 molecules-26-01204-f007:**
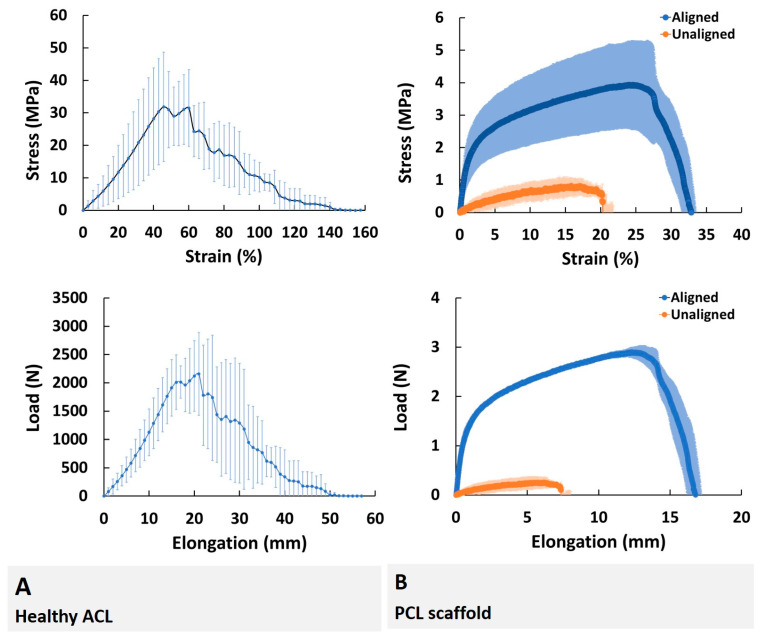
Mechanical properties of native ACL tissue and PCL scaffolds. (**A**) Healthy ACL tissue and (**B**) aligned and unaligned PCL scaffold with bimodal and unimodal fiber diameter distributions, respectively. Error bars represent SD.

**Figure 8 molecules-26-01204-f008:**
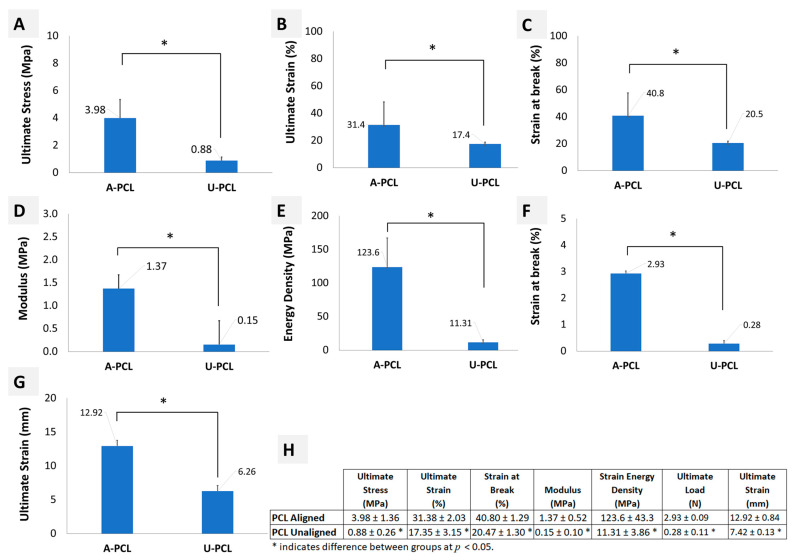
Mechanical properties of aligned (A-PCL) and unaligned (U-PCL) PCL scaffolds (**A**–**G**), and descriptive statistics (**H**). * Indicates significant difference at *p* < 0.05 determined using Student’s t-test. Error bars represent SD.

**Figure 9 molecules-26-01204-f009:**
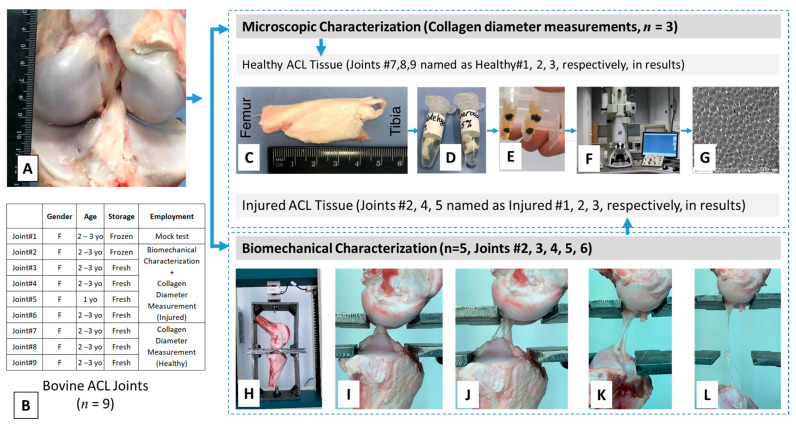
Experimental design of and representative images from native ACL harvesting and characterization. The ACL joint (**A**), demographics and processing of tissues before characterization (**B**), and procedure for transmission electron microscopy (TEM) characterization (**C**–**G**) and mechanical characterization (**H**–**L**).

**Table 1 molecules-26-01204-t001:** Typical composition of ACL tissue [9].

Cellular Materials (20%)	Extracellular Materials (80%)		
Fibroblasts	Water (60–80%)	Solids (20–40%)	
		Collagen 70–80%	Elastin
		Type I—90% Type III—10%	

**Table 2 molecules-26-01204-t002:** Mechanical properties of ACL tissue.

Species	Age/Weight	Cross-Head Speed (mm/min)	Modulus (N/mm)	Maximum Load (N)	Source
Human	22–35 years	20	242 ± 28	2160 ± 57	[13]
Sheep	9 months	5	144.97 ± 35.34	548.78 ± 41.44	[1]
Porcine	NR	5	43.5 ± 7.1	1055.5 ± 151.2	[14]
Rat	279 g	5	37.5 ± 11.5	47.8 ± 9.2	[15]
Sheep	4 months	6	136.3 ± 28.5	759.2 ± 114.1	[16]
Sheep	NR	60	44.5 ± 12.5	1531.3 ± 180.3	[17]
Bovine	12 months	500	204.1 ± 89.5	3317 ± 819	[18]
Bovine	3–7 years	60	577.3 ± 483.1 *	4372 ± 1485	[19]
Bovine	Mature	60	NR	4541 ± 1417	[20]

NR: not reported. * Recalculated from the source for unit consistency.

**Table 3 molecules-26-01204-t003:** Fibril diameter changes of different adult species upon injury.

Animal	Peak(s) for Healthy Tissue (nm)		Range Healthy (nm)	Peak for Injured Tissue (nm)	Range Injured (nm)	Reference
	Smaller	Larger				
Human ACL	~50	~120&150	20–200	NR	NR	[10]
Human ACL	75	NR	20–185	71	20–290	[25]
Rabbit ACL	~20	~250	10–320	NR	NR	[26]
Rabbit MCL	~40	~190	20–270	~50	40–70	[27]
Bovine ACL	~60	~125	40–250	NR	NR	[28]
Mouse PT	~45	~145	15–215	~45	15–125	[29]
Rat PT	~50	~210	20–380	~50&170	20–380	[30]
Mouse AT	~50	~170	10–320	NR	NR	[31]
Mouse FT	~60	~270	40–400	NR	NR	[22]

NR: not reported. MCL: Medial Collateral Ligament, PT: Patellar Tendon, AT: Achilles Tendon, FT: Flexor Tendon.

## Data Availability

Data is contained within the article or Appendix A.

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
