# Peer review of "Change in Collagen Fibril Diameter Distribution of Bovine Anterior Cruciate Ligament upon Injury Can Be Mimicked in a Nanostructured Scaffold"

_molecules, 2021, doi:10.3390/molecules26051204_

Round 1
Reviewer 1 Report
The Authors have corrected the manuscript as suggested by the Reviewer, significantly improving the quality of the manuscript. Just double check the figure 1 in which seams there some squares around the figure. The Reviewer suggest the publication after this minor revision.
Reviewer 2 Report
This study aimed at developing PCL scaffolds for Anterior Cruciate Ligament (ACL) related injuries which analyze the diameter of collagen fibrils of native bovine ACL tissue. In my opinion, you’ve done a sufficient job to write this research paper but not sufficiently sound for publication in the journal unless a major revision is performed. The following suggestions should be considered:
1. The introduction is not concise enough to highlight the subject and there are more than ten paragraphs in the introduction, please write less about the background of Anterior Cruciate Ligament injuries.
2. It is said in the manuscript that PCL was selected as the biomaterial for fabricating scaffolds due to the ease of electrospinability, in my opinion the reason is inadequate since poly(lactic-co-glycolic acid) and poly(lactic acid) are also accessible for electrospinability.
3. Your English writing skill should be enhanced since there are several grammar mistakes.
4. TEM images in Figure 2 are blurred, especially for B2 and B3, please provide high resolution pictures for reader’s convenience.
5. What is the best fibril diameter distribution? Does the aligned and unaligned PCL scaffold fiber diameter distribution qualitatively similar to the healthy and injured bovine ACL tissue mean good fiber diameter distribution? Please make a comparison chart to illustrate this.
6. Biomechanical properties of healthy ACL tissue and PCL scaffold were tested by tensile properties, how about the compressive properties? For this section, some recent research may give the authors some interesting insights and could be discussed, such as: Bioactive Materials, 2021, 6:490-502, ACS applied materials & interfaces, 2017, 9(5): 4890-4897.
7. As reported by some studies, the degradation properties is of great importance for PCL scaffolds, please supplement the tests (such as BPS immersion) to evaluate the degradation properties.
Round 2
Reviewer 2 Report
The paper can be accepted!
This manuscript is a resubmission of an earlier submission. The following is a list of the peer review reports and author responses from that submission.
Round 1
Reviewer 1 Report
In the paper “Change in collagen fibril diameter distribution of bovine Anterior Cruciate Ligament upon injury can be mimicked in a nanostructured scaffold” the Authors have investigated the morphological differences (via TEM) in the fiber diameters between healthy and injured bovine ACL ligaments. The mechanical properties of healthy ACL ligaments were tested with a tensile test. Later they realized different co-electrospun PCL mats (random and aligned) obtaining nanofibers with different ranges of diameters. They studied the nanofiber diameters distribution (SEM images) and their mechanical properties using a tensile test. The work is interesting but quite simplified and prelaminar, especially in the nanofibers characterization. Moreover, data need to be improved with additional tests and corrected in some parts. For this reason, the Reviewer suggests a resubmission after major revisions.
Introduction
1) Table 2: the term strain rate seems to be used incorrectly. In fact, the term strain rate is defined as the percentage of strain of the specimen per unit of time, considering the initial gauge length of the specimen itself. Conversely in this table the crosshead speed of the machine is reported. This term does not give any information about the strain applied to the samples because it does not consider the gauge length of the specimen tested. Please correct the table adequately.
2) Line 117: the paper focuses on electrospun scaffolds for tendon and ligament tissue engineering please add the following reference to give a comprehensive overview of the topic:
Sensini A, Cristofolini L. Biofabrication of Electrospun Scaffolds for the Regeneration of Tendons and Ligaments. Materials (Basel). 2018 Oct 12;11(10):1963. doi: 10.3390/ma11101963.
3) Lines 121-123: please remove the sentence in brackets (Dated 29 July 2019, keywords: 122 injury of bovine ACL, tensile properties of the cow ACL, collagen fiber diameter distribution of 123 bovine ACL, databases: scholar.google.com, NU library e-resources). These informations do not add any relevant information to the study.
Results
4) Figure 2: the TEM images seem to be of very low quality and in the case of figures B2 and B3 not in focus. The lower banner in the images (i.e. university logo and image parameters) does not add any information to the images because they are not visible. Please remove these lines from the images. In addition, the magnification of the images should be declared in the caption or in the materials section. Moreover, make a diameter distribution analysis with only 3 images for the sample family in the Reviewer’s opinion is not enough. Please analyze more images, also in different parts of the same specimen, to make the analysis more consistent. The magnification of images B1, B2 and B3 seems to be very different and the comparison risk to be affected by some artifacts. Please use the same magnification for all the images.
5) Figure 3: please remove the black banner from the figures. It is not readable and for this reason they do not add any additional information to the images. Maybe add the most important informations in the materials and methods section with also the magnification of the images. Similarly, in comment 4, increase the number of images analyzed (and also more different samples) to make the analysis more consistent. Moreover, the aligned nanofibers seem to not be perfectly aligned, an orientation analysis should be very important to quantify this point.
6) Lines 215-217: the Authors state that “the PCL scaffolds exhibited a typical tri-phase pattern with an initial toe region, followed by linear region, and finally the yield region”. The load cell used in the study (1 kN) is definitely off-scale for scaffolds that have a failure load from 0.28 up to 3 N. No informations are reported about the class of the cell and the acquisition system and the data postprocessing. Please integrate the draft in the materials section. Moreover, the toe region is not visible in the aligned mats (this can support my previous point in which I suggest an orientation analysis to confirm the axial alignment of the nanofibers) and for sure not visible in the random ones. This for the isotropic arrangement of the random fibers that strain in an isotropic way the mats, not permitting to show the toe region.
Discussion
7) Lines 285-287: the Authors state “Although electrospinning is a versatile technique to fabricate nanofibers, there are always limitations associated with this technique one of which is its incapacity to fabricate fibers with diameters as small as native ACL tissue”. However, there are a lot of works in literature that have demonstrated that electrospinning can reach such nanometric diameters. Please correct the sentence. See for example:
Jian S. et al, Nanofibers with diameter below one nanometer from electrospinning, RSC Adv., 2018, 8, 4794–4802 DOI: 10.1039/c7ra13444d
Sensini, A., et al. Tendon Fascicle-Inspired Nanofibrous Scaffold of Polylactic acid/Collagen with Enhanced 3D-Structure and Biomechanical Properties. Sci Rep 8, 17167 (2018). https://doi.org/10.1038/s41598-018-35536-8
Sensini A, et al. Morphologically bioinspired hierarchical nylon 6,6 electrospun assembly recreating the structure and performance of tendons and ligaments. Med Eng Phys. 2019 Sep;71:79-90. doi: 10.1016/j.medengphy.2019.06.019
Sensini A. et al., Hierarchical electrospun tendon‐ligament bioinspired scaffolds induce changes in fibroblasts morphology under static and dynamic conditions. Journal of Microscopy, 277: 160-169 (2020). https://doi.org/10.1111/jmi.12827
8) In the Reviewer’s opinion, also the mechanical properties of the injured ligaments should be tested to complete the study.
9) Lines 302-304: the Authors state “we preferred a deformation rate of 5mm/min”. This is not a strain rate but the crosshead speed. Please clarify this point and correct the sentence.
Materials and methods
10) Line 404: please report the diameter of the drum collector used and the related peripheral speed. Clearly state if the scaffolds were produced at room temperature and humidity. Moreover, no informations about the shape of the samples produced.
11) Line 418: please add the following parameters for the mechanical tests: the number of specimens used, the shape of the scaffolds tested, class pf the loading cells, the method to calculate the diameter of the specimens, gauge length of the specimens, how the stress was calculated. Moreover, a dedicated figure showing how the mechanical properties were obtained from the mechanical data would be very useful.
Conclusions
In the Reviewer’s opinion, this study is very prelaminar and other data need to be collected in the future to make these scaffolds suitable for ligament tissue engineering such as in vitro/in vivo tests, improvements of the mechanical properties and upscale o the hierarchical complexity of the scaffolds morphology. Please correct the conclusions in terms of a very preliminary study.
Reviewer 2 Report
In this manuscript, the fiber diameter changes after tissue injury in bovine anterior cruciate ligament (ACL) were investigated. The PCL scaffolds with bimodal and unimodal fiber range were prepared and their mechanical properties were tested to be compared with ACL. This is a very interesting and original point of view in tisse injury. These experimental results can provide evidence for the future research of ACL tissue repair. However, there are still some points that need to be modified:
- It might be clearer if a schematic representation of an ACL injury can be given in the introduction part.
- Some problems of ACL tissue repair are mentioned in the introduction part, but the application value of the research results to ACL tissue repair is not fully explained, which should be supplemented in the discussion part.
- Figure 1 is put at the end, and the order of the pictures needs to be checked and modified.
- Figure 7 is mentioned in line 232, but it is not found in the manuscript.
- The full name of "PCL" is not given in this paper.
- The full name of “ACL” in line 42 should be put in the first sentence of this paragraph.
- "2" in "mm2" of line 71 should be superscript.
Round 2
Reviewer 1 Report
The Authors have answered and integrated most of the Reviewer’s comments, however some data, in the Reviewer’s opinion, need to be improved to confer more consistence to the paper especially in the electrospun mats characterizations and in the biomechanical tests. For these reasons, I suggest a resubmission after major revisions.
General Comment
In the Reviewer’s opinion figure 1 seems to be quite hasty. Please redraw the figure (maybe using a graphic software) to increase its readability.
Results
1) Please add to the paper the orientation analysis of the nanofibers after a comparison of repeated measures to several images (at least 5 images for each sample category). Moreover, use mean and standard deviation to show the results of the frequency for each angle analyzed. This analysis is particularly fast and it will help the Reader to better focalize the repeatability and the variability in the manufacturing process. This analysis is a gold standard for the electrospun scaffolds for ligament applications. To have a possible suggestion on the workflow of this SEM-based analysis please consider the following paper if it could be useful:
SENSINI, A., CRISTOFOLINI, L., FOCARETE, M., BELCARI, J., ZUCCHELLI, A., KAO, A. and TOZZI, G. (2018), High‐resolution x‐ray tomographic morphological characterisation of electrospun nanofibrous bundles for tendon and ligament regeneration and replacement. Journal of Microscopy, 272: 196-206. https://doi.org/10.1111/jmi.12720
2) Mechanical tests: please report the data in the table inside of figure 6 also as histograms clearly showing the statistical differences between the different categories with the dedicated asterisks. This will help the Reader to easily read the data.
Materials and methods
3) Biomechanical characterization of ACL samples: Please report the gauge length of the ACLs tested and the cross-sectional diameters with mean and standard deviation and the instrumentation used (model, producer) and eventually the procedure used to calculate them.
4) Line 440: The Authors report a thickness of the mats raging between 50 an 100 micron. In the Reviewer’s opinion this data are too vague also considering the different nature of the samples tested (i.e. random mats = high thickness of the specimens according to the higher amount of air; aligned mats = less thickness according to the higher compactness of the specimens). Please clearly explain the number of repeated measures for each specimen and the instrumentation used. Please report the data in terms of mean and standard deviation.
5) Line 460: the Authors state that the mechanical tests on the electrospun mats were performed on (n=3) specimens per samples category. In the Reviewer’s opinion this is a too low number of specimens of each category. It is well known in fact that the electrospun materials are always characterized by a quite high variability both for the production methods and electrospinning parameters. The Reviewer suggest to integrate the test with at least 5-10 specimens for sample category (also to increase the statistic power of the analysis).
6) The Reviewer agree with the Authors that knowing the rpm and the collector diameter is easy to obtain the peripheral speed. However, the Reviewer suggests to report the peripheral speed data to allow the inexpert reader to orient himself more quickly in reading the paper.
7) Reviewer’s comment:”…Moreover, a dedicated figure showing how the mechanical properties were obtained from the mechanical data would be very useful.”
Authors reply: The data directly obtained from the device is given in Figure 6 together with its conversion to stress-strain plot. In Fig 6, elongation is the difference between the length at any time and the gauge length. We hope that this explains the method we used to obtain our mechanical properties and how we reported them.
Reviewer’s reply: thanks to the Authors for the explanation. However, in the Reviewers opinion this is a real critic point for the mechanical characterization of a ligament-inspired scaffold. Please clearly declare in the text the criterion you used to calculate the elastic modulus: what sample you considered as starting point of the stress strain curve? How you have defined the yielding point? Have you calculated also a limit of proportionality? If the toe region of the electrospun mats is not present, as it seems from the figures, please clearly declare it in the draft.
Discussion
8) Lines 327-333: In the Reviewer’s opinion these sentences do not clearly represent the state of the art of electrospun scaffolds for ligament applications. The possibility to increase the nanofiber range of diameters in such scaffolds is a part of the problem. In fact, at the current stage the scaffolds proposed in the manuscript are not able, at the moment, to reproduce the hierarchical structure of ligaments, and this point is confirmed from the not biomimetic mechanical properties. Moreover, the Authors have tested only mats of nanofibers, and this level of hierarchical aggregation is far from a biomimetic morphology. There are and extensive literature that proposes a lot of hierarchical biomimetic scaffolds for these applications. Please rephrase the sentences speaking only in term of really preliminary results in terms nanofibers morphology.
